# Cross-Quality Few-Shot Transfer for Alloy Yield Strength Prediction:
# A New Materials Science Benchmark and
# A Sparsity-Oriented Optimization Framework

Xuxi Chen[1], Tianlong Chen[1], Everardo Yeriel Olivares[1], Kate Elder[2], Scott K. McCall[2],
Aurelien Pierre Philippe Perron[2], Joseph T. McKeown[2], Bhavya Kailkhura[2],
Zhangyang Wang[1], Brian Gallagher[2]
[1]University of Texas at Austin [2]Lawrence Livermore National Laboratory
{xxchen,tianlong.chen,everolivares,atlaswang}@utexas.edu
{elder7,mccall10,perron1,mckeown3,kailkhura1,gallagher23}@llnl.gov

Discovering high-entropy alloys (HEAs) with high yield strength (YS) is crucial in materials science. However, the YS can only be accurately measured by expensive and time-consuming experiments, hence cannot be acquired at scale. Learning-based methods could facilitate the discovery, but the lack of a comprehensive dataset on HEA YS has created barriers. We present **X-Yield**, a materials science benchmark with 240 experimentally measured ("high-quality") and over 100,000 simulated ("low-quality") HEA YS data. Due to the scarcity of experimental results and the quality gap with simulated data, existing transfer learning methods cannot generalize well on our dataset. We address this *cross-quality few-shot transfer* problem by **leveraging model sparsification "twice"** — as a noise-robust feature regularizer at the pre-training stage, and as a data-efficient regularizer at the transfer stage. While the workflow already performs decently with sparsity patterns tuned independently for either stage, we propose a bi-level optimization framework termed *Bi-RPT*, that jointly learns optimal masks and allocates sparsity for both stages. The effectiveness of *Bi-RPT* is validated through experiments on **X-Yield**, alongside other testbeds. Specifically, we achieve a reduction of 8.9-19.8% in test MSE and a gain of 0.98-1.53% in test accuracy, using only 5-10% of the hard-to-generate real experimental data. The codes are available in https://github.com/VITA-Group/Bi-RPT.

## 1. Introduction

Machine learning (ML) methods have recently demonstrated great promise in the field of materials science. In this paper, we focus on ML-assisted high-entropy alloy (HEA) discovery [1]. HEAs have promising properties that traditional alloys do not hold, such as extraordinary mechanical performance at high temperatures, making them well-suited for improved materials applications. One particularly important mechanical property for customized HEA design is the Yield Strength (YS), which represents the maximum stress a material can withstand before it begins to deform plastically. However, in order to accurately measure the YS of specific HEAs, expensive scientific experiments must be performed for each alloy, often involving time-consuming and hard-to-create experimental conditions, especially at high temperatures (mainly caused by difficulties with oxidation control). Consequently, most of the experiments are performed at room temperature.

However, even for the least challenging YS experiments at room temperature, a team of domain experts typically requires two to four weeks to complete the entire process, including sample preparation (*i.e.,* alloy synthesis, homogenization heat treatment, and machining) and mechanical testing. Hence, it is extremely challenging and expensive to acquire YS measurements from such "high-quality" experiments at scale (*i.e.,* for multiple alloy compositions) and across relevant temperature domains (ranging from room temperature to high temperature).

First Conference on Parsimony and Learning (CPAL 2024).

Similar to the trends in computer vision fields [2], recent efforts attempted to mitigate the scarcity of real-world measurements by employing machine learning-based predictors that can directly predict their YS from the alloy inputs [3]; and such predictors could be trained using simulated data. Indeed, materials science applications often benefit from developed simulation models, such as the one proposed by Maresca et al. for YS prediction of single-phase body-centered cubic (BCC) HEAs [4]. However, a domain gap exists between the simulated data and the "ground-truth" experimental data. This gap is often the result of simplifications in the simulation modeling process that are necessary, but may not fully capture the complexity of the real-world system. For example, the YS of a material can exhibit significant variability depending on factors such as processing and testing conditions, as well as grain size and texture [5, 6]. However, most simulation models commonly rely on intrinsic alloy properties and do not incorporate variations resulting from different experimental conditions. The lack of public datasets in this field also renders it difficult to benchmark ML models' progress.

In this paper, we curate a materials science benchmark, called **X-Yield**, that for the first time combines existing experimental data with simulation data to address the problem of predicting YS in refractory HEAs. The former were carefully screened and curated from an open-source materials database [7] by domain experts while the latter were predicted by a theoretical yield strength model [4]. While the use of *"high-quality"* experimental data is always preferred, it is impractical to generate large quantities of data, especially for capturing YS at elevated temperatures. Thus, simulation data can be acquired in massive quantities to bridge the gap, despite their relatively *"low quality"* due to inherent model misspecification or simplification. The low-quality simulation data used in **X-Yield** was selected to represent ternary-septenary systems from an eleven-element palette consisting of mostly refractory elements (Al-Cr-Fe-Hf-Mo-Nb-Ta-Ti-V-W-Zr). Although there are experimental databases available [7] and predictive models for high-temperature YS in HEAs [4], to the best of our knowledge, this is the first multi-fidelity dataset in the public domain that combines real experimental measurements and large quantities (over 100K) of simulation data for mechanical property (*i.e.*, YS) prediction in HEAs. This specialized dataset should be able to predict high-temperature YS across a broad range of refractory HEAs. The predictions of this model could be used to pinpoint which alloys are the strongest at elevated temperatures, allowing experiments to focus on pre-sorted candidates for future study and eliminating the need to spend several weeks testing a candidate without promise.

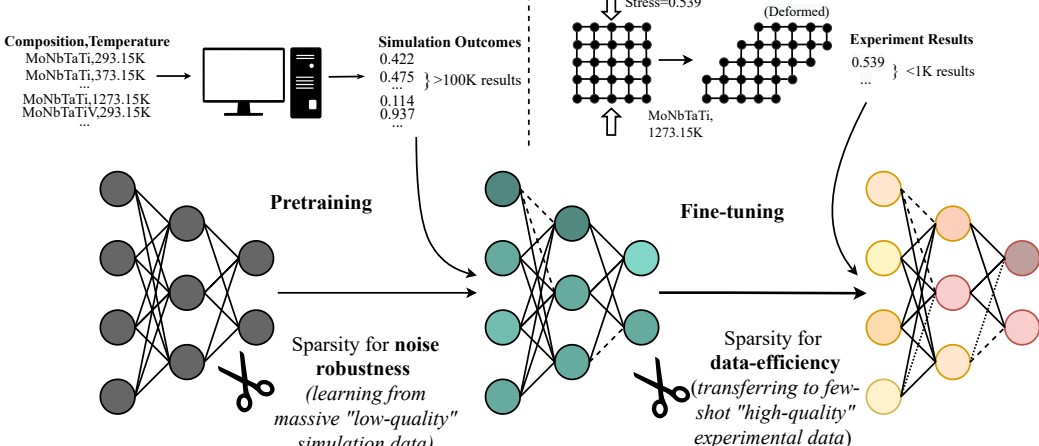

Figure 1: Proposed two-stage workflow. The HEA yield strength prediction model is first pre-trained on massive "low-quality" simulation data, and is then fine-tuned/transferred on few-shot "high-quality" experimental data to optimize its prediction in this target domain. Note that the tool of sparsity will be leveraged in both pre-training and fine-tuning stages, for the purposes of gaining noise robustness/transferablity and enhancing data efficiency, respectively.

The new **X-Yield** benchmark is set to facilitate ML for HEA yield strength prediction, but learning from such a multi-fidelity dataset is highly challenging. To address this challenge, we conceptualize a **cross-quality few-shot transfer workflow**. This involves pre-training the prediction model

on the data-rich yet "low-quality" source domain (simulated data), and then fine-tuning the model towards the data-scarce yet "high-quality" target domain (experimental data). However, the efficacy of this vanilla workflow is hindered by two major challenges: a significant quality gap between source and target domains, and an extreme data scarcity of target data. Inspired by the recent success of sparsity regularizers, we propose to incorporate sparsity to regularize both stages: sparsifying pre-training to improve the robustness and cross-domain transferability of learned features [8–13], and sparsifying fine-tuning to overcome data shortfalls [14–16]. We demonstrate proof-of-concept experiments that even the simplest magnitude-based weight pruning could play effective regularization roles in our workflow. Furthermore, to avoid the ad-hoc two-step pruning as well as trial-and-error sparsity ratio selection at either stage, we propose a novel integrated optimization framework termed *Bi-Level Regularized Pre-training and Transfer* (***Bi-RPT***), that jointly learns optimal sparse masks and automatically allocates sparsity levels for both stages. Our main contributions are summarized as follows:

- **Dataset:** We present **X-Yield**, the first public multi-quality materials science benchmark for refractory HEA yield strength prediction, containing alloys compositions, processing temperatures, and yield strength. Specifically, the experimental YS of 240 HEAs are reported, while YS of the remaining samples (over 100K) are calculated by simulations.

- **Methodology:** we formulate a cross-quality few-shot transfer workflow that can jointly exploit the simulated and experimental data for accurate predictions, and we innovate to leverage sparsity for addressing both the simulated/experimental domain gap and the scarcity of experimental data. While ad-hoc magnitude-based weight pruning is already found to be helpful, we further formulate an integrated bi-level optimization framework called ***Bi-RPT*** to automate the optimal sparse mask generation and sparsity ratio allocation at both pre-training and fine-tuning stages. The overall pipeline is described in Figure 1.

- **Results:** Extensive experiments show that *Bi-RPT* can boost performance on the **X-Yield** benchmark alongside other synthesized testbeds. In particular, for the YS regression task, we achieve a reduction of 19-38% on the test mean squared error by using only 5-10% of the available experimental data. For the YS classification task, we achieve 0.98-1.53% improvement in test accuracy.

## 2. Related Works

### 2.1. Machine Learning in Materials Research

ML has been applied to solve a wide range of problems in materials science ranging from the fields of inorganic chemistry [17] to sustainability [18], and metallurgy [19], with the typical purposes to predict materials properties and accelerate simulations [20]. In general, ML techniques significantly reduce computational time compared to traditional materials science methods and are typically fast to develop [21]. More recently, deep learning has been successfully applied to problems in the field of HEAs, in particular to predict phase formation [22, 23]. Deep learning provides significant increases in speed compared to CALculation of PHAse Diagrams (CALPHAD) [24], density functional theory [25], and molecular dynamics methods [26] commonly used in materials science. Deep learning has also been applied to predict other properties such crystal structure, elastic constants [27], hardness [28], and (most relevant to us) yield strength of refractory HEAs [3, 27]. However, previous deep learning efforts for YS prediction are restricted to the development of specific alloys [28–30] and many studies use only a small experimental dataset for prediction [31]. In contrast, this work presents a general multi-fidelity ML model for predicting YS of HEAs at scale.

### 2.2. Sparsity Regularization in Deep Learning

Sparsity or pruning was traditionally treated as a mainstream model compression approach in deep learning [32]. Recently, sparse regularizers have been increasingly used to enhance deep model robustness to various noise, malicious attacks, and distribution shifts. Previous work studied the intrinsic relationship between pruning and adversarial robustness [8, 9, 33]. Other recent work [13] comprehensively demonstrated the benefit of model sparsification to improve robustness to distributional shifts [34, 35]. Sparse regularizers also exhibit promise in improving data efficiency.

For example, Zheng et al. and Liu et al. [14, 36] proposed to learn model pruning strategies for few-shot learning and Tian et al [37] combined model sparsification with meta-learning to improve few-shot performance. Sparse regularizers have even been proven effective beyond few-shot image classification, such as enhancing the data efficiency in image generation [15].

### 2.3. Bi-Level Optimization

Bi-level optimization is a hierarchical framework where the variables in the *upper-level* optimization problem are dependent on the *lower-level* problem. Finn et al. and Rajeswaran et al. [38, 39] formulated the meta-learning problem in the form of bi-level optimization, and solve it by using first-order approximations. Other applications of bi-level optimization include data and label poisoning [40, 41], and adversarial training [42]. In this work, we utilize bi-level optimization to formulate a two-stage workflow with sparsification and find each stage's optimal weights and sparse masks while considering their sequential dependency.

## 3. X-Yield: A new benchmark for refractory HEA yield strength prediction

**Overview.** Conventional alloys typically have one principal element with small amounts of other elements added to improve material properties [43] while HEAs can have multiple principal elements. The discovery of HEAs opened the door to a significantly larger design space to explore, most of which has yet to be examined [44]. To address the task of using ML to predict HEA yield strength, we focus on the sub-field of refractory HEAs (RHEAs). These materials have been demonstrated to maintain excellent mechanical properties (*i.e.*, YS) at high temperatures [45], making them ideal candidates for hypersonics and aerospace industry applications. Prior work adopting ML to predict RHEA properties either uses solely experimental data [31], or restricts predictions to smaller composition space and single temperature [30] or specific alloys such as MoNbTaTiW at multiple temperatures [3]. Hence, a generalizable ML prediction model for a broad range of RHEAs across temperature domains is still absent. As mentioned earlier, it is impractical to generate high quantities of experimental data, especially for capturing YS at elevated temperatures. There are also challenges specific to high-temperature measurements such as controlling oxidation, confirming the heating profile and gradient within the samples, and use of more challenging experimental techniques (crosshead displacement) than those at lower temperatures (extensometers).

This work develops **X-Yield**, the first publicly available, multi-fidelity dataset consisting of over 100K low-quality simulated points and 240 experimental data points to explore the RHEA design space. In this study alone, the entire composition space of all alloys containing between ternary-septenary systems from the Al-Cr-Fe-Hf-Mo-Nb-Ta-Ti-V-W-Zr family is examined. Since obtaining real high-temperature YS data is challenging, a majority of the experimental YS data in the literature was taken close to room temperature [7] even though there is more interest in RHEA properties at high temperature [44]. **X-Yield** can be used to train a multi-fidelity ML model to predict high-temperature YS for a broad range of RHEAs. The combination of high-temperature YS from the simulated dataset and experimental input can generate an ML model to accurately and efficiently predict high-temperature YS of alloys not included in the training set.

**Dataset Construction.** The YS of the simulation data was predicted using the analytic and parameter-free mechanistic yield strength model developed by Maresca et al. [4]. This model describes body-centered cubic (BCC) multi-principal element alloy (MPEA) solid solution strengthening associated with edge dislocations, in terms of elemental atomic volumes and elastic moduli. The YS was predicted for all ternary (1% increments), quaternary (1% increments), quinary (5% increments), senary (5% increments), and septenary (5% increments) alloys from the Al-Cr-Fe-Hf-Mo-Nb-Ta-Ti-V-W-Zr element family at temperatures between 300K-2500K in increments of 100K. This resulted in over three billion data points of which approximately 100,000 were randomly selected for inclusion in this study. Note that even this advanced simulation model suffers from notable oversimplification and data quality issues. For example, the phase stability and dislocation character were not used to filter alloys in the study and the model may overpredict the YS of alloys with non-BCC phases and underpredict those with different dislocation character (*e.g.* screw).

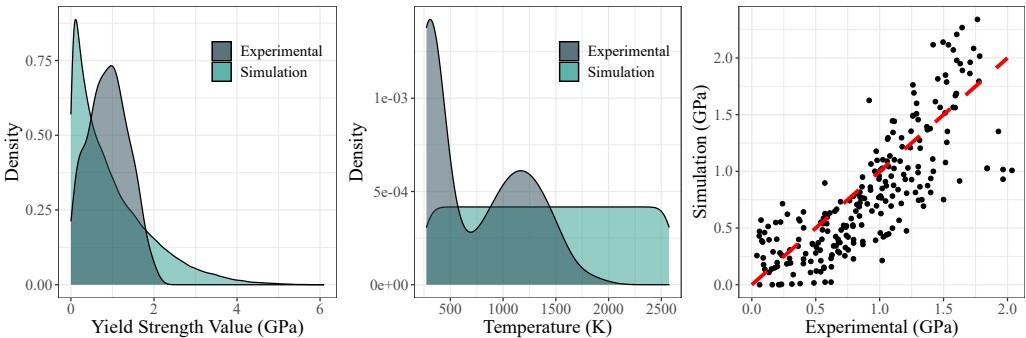

Figure 2: Left: distribution of YS; Middle: distribution of temperature; Right: pairwise visualization of YS.

The high-quality experimental dataset was carefully filtered and curated from the database generated by Borg et al. [7] consisting of mechanical property information (*e.g.*, YS) for MPEAs. All extracted data points consisted solely of alloys based on the elements from the above element family exhibiting only BCC phases, and containing a YS value at temperatures higher than 20°C.

**Dataset Characteristics and "Quality Gap".** As depicted in Figure 2, the simulation and experimental YS have different distributions. In the low-quality simulation data, a considerable portion of YS values is greater than 2 GPa, while the experimental data contains almost no YS points beyond 2 GPa (with only one exception). The distribution of the simulated YS is also significantly more skewed than the experimental one. The testing temperature distributions also differ significantly, with the simulation data presenting an near-uniform pattern while the experimental temperature distribution is bimodal with a dominant low-end peak (*i.e.*, reflecting the difficulty to acquire experimental data at high temperature compared to room temperature). These observations showcase the domain shifts or a "quality gap" between simulations and experiments. As a result, the pairwise visualization of the YS on the 240 high-quality experimental samples suggests a substantial deviation between simulated and experimentally observed yield strength.

## 4. Cross-Quality Few-shot Transfer: A Two-Stage Workflow aided by Sparsity (Twice)

In this section, we will begin by introducing the basic two-stage workflow. Building on this foundation, we propose a novel sparsification framework consisting of two approaches: a vanilla approach called "Hand-Tune" and an improved principled framework named "*Bi-RPT*".

**Basic Two-Stage Workflow: Pre-training then Fine-tuning** Let us denote the high-quality target domain (experimental data) by $\mathcal{D}_t$, and the low-quality source domain (simulated data) by $\mathcal{D}_s$. Our goal is to learn a generalizable predictor over $\mathcal{D}_t$ while leveraging the aid of $\mathcal{D}_s$. One naive idea is to simply combine the two data domains and jointly train a supervised model. However, the large domain gap between $\mathcal{D}_s$ and $\mathcal{D}_t$, as well as the sample scarcity in $\mathcal{D}_t$, result in a poorly trained fit to $\mathcal{D}_t$. Instead, we propose to formulate our workflow as a two-stage pipeline: first pre-training a model on $\mathcal{D}_s$, and then fine-tuning to optimize the prediction over $\mathcal{D}_t$.

**Incorporating Bi-Stage Sparsity: A Vanilla Approach.** Despite the usefulness of features learned from $\mathcal{D}_s$, it is inevitable that they suffer from domain gap and noise when applied to $\mathcal{D}_t$. Furthermore, the scarcity of data in $\mathcal{D}_t$ poses another challenge. In light of recent successes achieved through the use of sparse regularizers in enhancing both robustness/transferability and data efficiency, we attempt to incorporate sparsity into both stages to address the two-fold challenges.

We first prove our concepts by proposing a vanilla ad-hoc approach, which we refer to as *Hand-Tune*. Starting from pre-training over $\mathcal{D}_s$, we perform the standard iterative magnitude pruning (IMP) [46] during pre-training. In particular, we alternate between (re-)training and pruning; each time, we prune the 20% smallest-magnitude weights from the existing non-zero weights by default and continue (re-)training the remaining non-zero weights. Such a "prune-and-retrain" routine is repeated for $N_s$ rounds to obtain the final sparse mask $\boldsymbol{m}_s$ (1 denotes the element to be non-zero and 0 to be pruned) associated with the pretrained model weight. Then, we move on to fine-tuning

over $\mathcal{D}_t$, and start another round of IMP on top of the pre-trained model: note that this second-stage IMP continues only on the subset of current non-zero weights, *i.e.*, the 1-valued regions in $\boldsymbol{m}_s$. IMP in fine-tuning repeats another $N_t$ round (with the identical protocol as the first stage), yielding another sparse mask $\boldsymbol{m}_t$. The final model uses the joint sparse mask $\boldsymbol{m}_s \odot \boldsymbol{m}_t$ where $\odot$ represents the point-wise product.

Hereby, $N_s$ and $N_t$ are hyperparameters that control the sparsity allocation between two stages. Intuitively, while certain sparsity may contribute to noise resilience, an overly large $N_s$ will cause the pre-trained model to be over-sparsified, limiting its capacity to learn sufficiently informative and transferable features. Fine-tuning has a similar trade-off. Therefore, $N_s$ and $N_t$ have to be manually tuned for the two-stage workflow to achieve good performance (see Appendix B.2).

**Principled Bi-Stage Sparsity Integration with *Bi-RPT*.** The *Hand-Tune* method has some notable drawbacks. Firstly, it uses weight magnitude information to remove weight elements, which is not explicitly driven by the task. Secondly, the two sparse masks $\boldsymbol{m}_s$ and $\boldsymbol{m}_t$ are determined sequentially rather than jointly optimized. As a result, learning $\boldsymbol{m}_t$ may be affected by any artifact in learning $\boldsymbol{m}_s$. Finally, the sparsity ratios assigned in both stages, controlled by $N_s$ and $N_t$, require manual tuning without any clear understanding beyond an exhaustive hyperparameter search.

We, therefore, devise a more principled framework that can jointly learn the optimal sparse masks as well as sparsity allocations for both stages, termed *Bi-Level Regularized Pre-training and Transfer* (***Bi-RPT***). The corresponding optimization problem can be expressed as follows:

$$\min_{\boldsymbol{\theta},\boldsymbol{m}_s,\boldsymbol{m}_t} \mathbb{E}_{(\boldsymbol{x}_t,y_t)\sim\mathcal{D}_t} [\mathcal{L}_t((\boldsymbol{m}_s \odot \boldsymbol{m}_t) \odot \boldsymbol{\theta},\boldsymbol{x}_t,y_t|\boldsymbol{\theta}^*,\boldsymbol{m}_s^*)] + \gamma\mathcal{R}(\boldsymbol{m}_s^* \odot \boldsymbol{m}_t) \qquad (1)$$

$$\text{s.t. } \{\boldsymbol{\theta}^*,\boldsymbol{m}_s^*\} = \arg\min_{\boldsymbol{\theta},\boldsymbol{m}_s} \mathbb{E}_{(\boldsymbol{x}_s,y_s)\sim\mathcal{D}_s} \mathcal{L}_s(\boldsymbol{m}_s \odot \boldsymbol{\theta},\boldsymbol{x}_s,y_s), \qquad (2)$$

where $\gamma$ is a coefficient, $\mathcal{L}_s/\mathcal{L}_t$ represents the objective functions for the two stages, respectively, $\boldsymbol{\theta}$ represents the models' parameters, and $\mathcal{R}$ represents the sparsity regularizer. Seemingly complicated at the first glance, the bi-level optimization formulation of *Bi-RPT* actually admits a clear physics "workflow" interpretation. Let us start from the *lower-level* problem (2) which instantiates the sparsity regularized pre-training stage over $\mathcal{D}_s$: its outputs include the pre-trained weight $\boldsymbol{\theta}^*$ and the corresponding sparse mask $\boldsymbol{m}_s$. Then, the *upper-level* problem (1) depicts the sparsity regularized fine-tuning over $\mathcal{D}_t$, which inherits both $\boldsymbol{\theta}^*$ and $\boldsymbol{m}_s^*$ as its starting point. It continues to modify the weight as well as to evolve another sparse mask $\boldsymbol{m}_t$. Eventually, a sparsity-promoting function $\mathcal{R}$ enforces the total sparsity over the joint mask $\boldsymbol{m}_s \odot \boldsymbol{m}_t$, and the final model weights could be represented as $(\boldsymbol{m}_s \odot \boldsymbol{m}_t) \odot \boldsymbol{\theta}$.

Importantly, the lower- and upper-level problems in *Bi-RPT* are solved in an end-to-end manner, meaning that even the fine-tuning depends on $\boldsymbol{\theta}^*$ and $\boldsymbol{m}_s^*$, it can, in turn, provide feedbacks for adjusting the latter: hence a synergistic optimization is achieved between two stages. The sparse mask selection now directly hinges on the end task (target domain loss $\mathcal{L}_t$) rather than heuristics such as weight magnitudes. Lastly, the sparsity levels of $\boldsymbol{m}_s$ and $\boldsymbol{m}_t$ do not need to be separately designated nor manually controlled: we automatically learn the sparsity ratio allocation, under only the total sparsity regularizer $\mathcal{R}$.

To practically solve the bi-level optimization of *Bi-RPT*, we derive algorithms whose details can be found in Appendix A. For the sparsity regularizer $\mathcal{R}$, we adopt the smoothed $\ell_0$ term [47] to facilitate differentiable training: a gate function $g_\epsilon(\boldsymbol{x}) = \boldsymbol{x}^2/(\boldsymbol{x}^2 + \epsilon)$, whose outputs are almost binary when the $\epsilon$ is small, is used. In general, for the lower-level optimization problem, we update the model parameters $\boldsymbol{\theta}$ by gradients to minimize $\mathcal{L}_s$; for the upper-level optimization, we utilize the gradient unrolling to develop update rules for $\boldsymbol{\theta}$.

## 4.1. Proof-of-Concept Experiments on Image Data

To evaluate the effectiveness of our proposed *Hand-Tune* and *Bi-RPT* methods, we conduct proof-of-concept experiments on a synthesized testbed of image classification. We adopt two source-domain dataset options: ImageNet [48] and ImageNet-C [34], the latter more noisy and corrupted. Two

target-domain options are also accompanied: CUB-200 [49] and *CUB-200* (*10-shot*), the latter designed to be rigorously "few-shot" where each class has only 10 training samples. By using different combinations of $\mathcal{D}_s/\mathcal{D}_t$, we can conduct controlled experiments to evaluate the noise robustness and data efficiency of various algorithm options.

Several baselines are compared to *Hand-Tune* and *Bi-RPT*: (1) *Pretrain-and-transfer*: the basic workflow of pre-training on $\mathcal{D}_s$ followed by finetuning on $\mathcal{D}_t$, with no sparsity involved; (2) *Pretrain sparsity only /transfer sparsity only*: following our proposed pretrain-and-transfer workflow, but conducting IMP to only the pre-training/finetuning stage; (3) *No Pretraining*: directly training on $\mathcal{D}_t$ without using $\mathcal{D}_s$; (4) *Mix Training*: training one model on $\mathcal{D}_s$ and $\mathcal{D}_t$ combined. For methods involving IMP, we carefully select the sparsity ratio(s) by hands for either or both stages that result in the best generalization performance on $\mathcal{D}_t$, using grid search through cross-validation.

Table 1 reports the accuracies of all methods over various source/target combinations, on the same testing set of CUB-200 in Table 1. All methods use the same ResNet-18 backbone. We highlight several key observations: (1) incorporating $\mathcal{D}_s$ in general helps both CUB-200 and CUB-200 (10-shot), and the improvement margin is much more substantial for the few-shot case; (2) models trained by *Mix Training* fail to generalize on $\mathcal{D}_t$ - in fact even worse than *No Pretraining*, showcasing the negative influence of the quality gap; (3) in the same regime of pre-training then fine-tuning, adding appropriate sparsity helps, and two-stage sparsity can help more; (4) *Bi-RPT* consistently outperforms *Hand-Tune* (especially, very notably in few-shot cases), despite the best efforts in tuning the latter's hyperparameters. More observations and analysis can be found in Appendix B (Tables 4 - 7, and Figure 5): including but not limited to the backfiring effect of "over-sparsification" and the compound influence of per-stage IMP sparsity allocation in *Hand-Tune*. In Section B.8, we present additional experimental results that compare *Bi-RPT* with an advanced few-shot learning technique, TIM [50]. The results clearly demonstrate that *Bi-RPT* achieves superior performance.

Table 1: Experiments on image data: testing accuracy of fine-tuned ResNet-18 on CUB-200 / CUB-200 (10-shot) as $\mathcal{D}_t$, after pretraining on ImageNet and ImageNet-C as $\mathcal{D}_s$, respectively.

| $\mathcal{D}_t$ | Methods | Two-stage | $m_s$ | $m_t$ | $\mathcal{D}_s$ | |
| --- | --- | --- | --- | --- | --- | --- |
| | | | | | ImageNet | ImageNet-C |
| | No Pretraining | ✗ | ✗ | ✗ | 44.27% / 7.98% | |
| | Mix Training | ✗ | ✗ | ✗ | 30.88% / 6.72% | 27.32%/6.89% |
| CUB-200 | Pretrain-and-transfer | ✓ | ✗ | ✗ | 74.16% / 38.66% | 71.59% / 32.14% |
| / CUB-200 (10-shot) | Pretrain sparsity only | ✓ | ✓ | ✗ | 76.01% / 40.73% | 73.70% / 38.76% |
| | Transfer sparsity only | ✓ | ✗ | ✓ | 74.16% / 38.90% | 71.83% / 32.53% |
| | Hand-Tune | ✓ | ✓ | ✓ | 76.01% / 40.78% | 74.01% / 39.94% |
| | *Bi-RPT* | ✓ | ✓ | ✓ | **78.60% / 51.55%** | **76.29% / 47.01%** |

# 5. Main Experiments on the X-Yield Benchmark

## 5.1. Implementation Details

**Task Definition.** On **X-Yield**, the most natural task is predicting the yield strength (YS) of alloys (*i.e.*, regression) and calculating the error between the model prediction and the "ground-truth" experimental results. In addition to the regression task, we have formulated a surrogate classification task by categorizing the ground-truth YS into five bins based on their values: $[0, 0.5)$, $[0.5, 1)$, $[1,1.5)$, $[1.5,2)$, and $[2,\infty)$ in GPa. These categorical labels enable us to perform a classification task that complements the regression analysis.

**Data Representations.** We featurize each HEA by mapping its composition and temperature into a "pseudoimage" (please refer to Appendix B.3 and Figure 4). The pseudoimages have two channels. The first channel encodes the HEA's composition using a randomized periodic table structure [51]. The second channel embeds the HEA's temperature, which we convert from kelvin to a normalized temperature $T_{\text{norm}} = (K - 273.15)/2000$, and then include as a second channel in the pseudoimage.

**Architectures.** Our ML predictor utilizes a convolutional neural network structure, which is composed of three convolutional layers. Each layer has a kernel size of three and is followed by Batch

Normalization [52] and ReLU [53] activation. Additionally, we have appended a multi-layer perceptron to the convolutional neural network to generate the final prediction for both regression and classification tasks. The hyperparameters we use are shown in Sec B.1.

**Evaluation Metrics and Data Splits.** We evaluate each method in two ways. Besides the widely used 10-fold cross-validation, we explore two challenging *extreme few-shot* settings. Specifically, we randomly sampled 5% (and 10%) of experimental data from each alloy type (ternary, quaternary, quinary and senary) as our training set, while the remaining data served as the testing set. Note that these alloy "types" are distinct from the classification labels, which are described in the **Task Definition** subsection above. The end result is that we have only 23 (or 11) training samples and 217 (or 229) testing samples in the *extreme few-shot* settings. All the low-quality (simulated) data is used for pretraining where applicable. For the regression task, we report the lowest mean squared error (MSE) achieved on the testing samples. For the classification task, we measure model performance by reporting their accuracy on the testing split.

## 5.2. Main Results

**Classification and regression with extreme few-shot settings.** We first apply *Bi-RPT* to solve the regression and classification tasks under the two extreme few-shot settings where only 5% and 10% of experimental data are available, respectively. Table 2 shows that: (1) pretraining on simulation data can benefit the ML predictor consistently on both the regression (over 10% reduction in MSE) and classification (over 11% improvement in accuracy) tasks, especially when the data is more scarce; (2) the integration of sparsity into the pretraining and transfer workflow can further strengthen the predictor's generalization, improving accuracy by 0.98% and reducing MSE by 8.91% in the 10% experimental data case and improving accuracy by 1.53% and reducing MSE by 19.75% in the 5% experimental data case.

Table 2: Machine learning prediction performance on the test set of different splits of high-entropy alloy data. The experiments are repeated 10 times, and we report both the mean and the 95% confidence interval.

| Method | 10% train samples | | 5% train samples | |
|---|---|---|---|---|
| | Test MSE | Test Accuracy | Test MSE | Test Accuracy |
| No Pretraining | $0.114 \pm 0.007$ | $54.84 \pm 1.59\%$ | $0.212 \pm 0.041$ | $47.25 \pm 0.80\%$ |
| Pretrain-and-transfer | $0.101 \pm 0.001$ | $65.85 \pm 0.88\%$ | $0.162 \pm 0.002$ | $62.84 \pm 1.96\%$ |
| *Bi-RPT* | $\mathbf{0.092 \pm 0.011}$ | $\mathbf{66.83 \pm 1.41}\%$ | $\mathbf{0.130 \pm 0.006}$ | $\mathbf{64.37 \pm 1.10}\%$ |

**Additional classification and regression results.** Table 11 presents results for the slightly more "data-rich" 10-fold cross-validation setting. Here, we observe a similar trend: the bi-stage pretraining and transfer approach outperforms the single-stage training pipeline, and incorporating sparsity consistently provides improvement to the ML predictor, particularly in the regression case.

**Performance comparison on alloys at various temperatures.** Based on the trained model with 10% experimental data, we predict the YS of three alloys, MoNbTaTi, MoNbTaTiW and HfMoNbTaTiZr, at different temperatures. Table 3 shows the predicted YS of these three alloys using *Bi-RPT* and baselines. On the quinary and senary alloy systems, *Bi-RPT* shows exceptional accuracy in predicting the experimental YS, while predictions on the quaternary system exhibit great improvements compared to the initial simulation data. More scrutiny of those predictions reveals several findings that neatly align with materials science theory. For example, it is known that screw dislocations are more likely to be dominant than edge dislocations in MoNbTi and NbTaTi ternaries (shown from the ternary comparison in the Citrine database [7]). Thus it makes sense that the Maresca-Curtin model [4] based only on edge dislocations and ignoring the effect of screw dislocations (refer to Simulation data in Table 3) under-predicts the MoNbTaTi and MoNbTaTiW cases. However, our model seems to correctly pick up these differences thanks to the multi-fidelity data sources including both edge and screw dislocations effects and predicts a higher YS. As the effects of the MoNbTi and NbTaTi systems are more diluted in the quinary than the quaternary system, it is interesting to note that our model performs better for the quinary system (*i.e.*, less "quality gap" difference between the low and high fidelity data). Another example is that the Maresca-Curtin model over-predicts

HfMoNbTaTiZr at lower temperatures (300K∼900K) and under-predict YS at higher temperatures (1100K∼1300K) compared with experiments. These discrepancies are directly related to the theory of the model, which only consider the effect of edge dislocations. For these alloys, screw dislocations are assumed to dominate the YS behavior at low temperature (so the Maresca-Curtin model is not designed to perform well in this temperature range) while edge dislocation will take over at higher temperature (so the Maresca-Curtin model is better equipped to predict YS in this temperature range, while neglecting the potential remaining influence of screw dislocations and other mechanisms). Our predicted YS values are in a much better agreement with experiments for this senary system, highlighting that the complex underlying behavior of screw and edge dislocations (and other undefined mechanisms) as functions of alloy complexity and temperature and its impact on the resulting YS can be captured by the multi-fidelity ML models.

Table 3: Predicted YS of different alloys at different temperatures. Only 10% of the experimental data are used for fine-tuning. We compare the predicted YS generated by *Bi-RPT* with our "No Pretraining" (NP) and "Pretrain-and-transfer" (PT) baselines and the simulation. Best results (smallest errors) are marked in **bold**.

| Alloys | Temperature (K) | Predicted Yield Strength (GPa) | | | | Experimental (GPa) |
|---|---|---|---|---|---|---|
| | | *Bi-RPT* | NP | PT | Simulation | |
| MoNbTaTi | 293.15 | 1.078 | **1.170** | 1.062 | 0.475 | 1.210 |
| | 473.15 | **0.965** | 1.004 | 0.902 | 0.381 | 0.868 |
| | 673.15 | 0.746 | 0.772 | **0.731** | 0.282 | 0.685 |
| | 873.15 | 0.508 | 0.642 | **0.584** | 0.472 | 0.593 |
| | 1273.15 | 0.425 | **0.570** | 0.488 | 0.114 | 0.539 |
| MoNbTaTiW | 298.15 | **1.268** | 1.031 | 1.068 | 0.814 | 1.399 |
| | 873.15 | **0.677** | 0.569 | 0.607 | 0.372 | 0.689 |
| | 1073.15 | **0.618** | 0.520 | 0.523 | 0.294 | 0.674 |
| | 1273.15 | **0.536** | 0.530 | 0.486 | 0.232 | 0.620 |
| HfMoNbTaTiZr | 296.15 | **1.527** | 1.051 | 1.132 | 1.849 | 1.515 |
| | 873.15 | **0.861** | 0.556 | 0.685 | 1.178 | 0.973 |
| | 1073.15 | **0.762** | 0.536 | 0.612 | 0.516 | 0.791 |
| | 1273.15 | **0.662** | 0.563 | 0.573 | 0.421 | 0.753 |

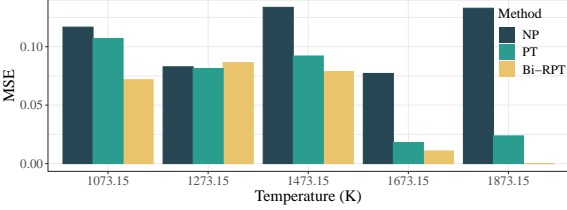

Figure 3: Prediction MSE at different temperatures. We compare the results of three methods: No Pretraining (NP), Pretrain-and-transfer (PT), and *Bi-RPT*.

**Prediction performance at high temperatures.** One critical task in the alloy design community is to find alloys capable of withstanding stress at high temperatures without deforming plastically. To assess the ability of *Bi-RPT* to aid the community in achieving this objective, we conducted a thorough analysis of its predictive performance under high-temperature conditions. We train our model with 10% of the experimental data, predict the YS for the remaining 90%, and compare the predictive quality of models at high temperatures in Figure 3. Our results indicate that *Bi-RPT* significantly outperforms the other models, particularly at temperatures greater than 1400K. These results suggest *Bi-RPT* could serve as a powerful tool for designing refractory HEAs with superior YS at elevated temperatures.

## 6. Conclusions

To address the important yet challenging problem of HEA yield strength prediction, we curated and released **X-Yield**, the first multi-fidelity HEA YS benchmark. To effectively leverage this benchmark, we also designed a two-stage cross-quality few-shot transfer workflow and proposed to use sparsity to tackle the two-fold challenges, i.e., low data quality during pretraining and data scarcity during fine-tuning. We formulate a principled bi-level optimization framework to automatically learn the optimal sparse masks and sparsity allocation between training stages. Extensive experiments on both image data testbeds and **X-Yield** demonstrate that *Bi-RPT* shows a substantial improvement over existing baselines. Moving forward, we are working closely with materials scientists to validate our ML prediction results based on their domain expertise, and the team has already identified some alloy candidates that appear promising for experimental validation.

# 7. Acknowledgments

This work was performed under the auspices of the U.S. Department of Energy by Lawrence Livermore National Laboratory under Contract DE-AC52-07NA27344 and was supported by the Laboratory Directed Research and Development (LDRD) program under project tracking code 22-SI-007 (LLNL-JRNL-848582).

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

## A. More Details on Methods

In this section, we present the technical details of our proposed method and framework ("Hand-Tune" and "Bi-RPT").

### A.1. Hand-Tune

The Hand-Tune algorithm determines the sparse masks for the two stages in an iterative manner, as outlined in Algorithm 1.

### A.2. *Bi-RPT*

We now build the techniques to solve the bi-level optimization problem formulated in *Bi-RPT*.

**Algorithm 1** Hand-Tune

---

**Input:** Initialization weights $\boldsymbol{\theta}_0$, low-quality pretraining dataset $\mathcal{D}_s$, high-quality fine-tuning dataset $\mathcal{D}_t$, number of IMP rounds $N_s$ for the pretraining stage and $N_t$ for the fine-tuning stage.
**Output:** the trained weights $\boldsymbol{\theta}^*$, the sparse mask $\boldsymbol{m}_s$ for the pretraining stage, and the sparse mask $\boldsymbol{m}_t$ for the fine-tuning stage.
Initialize the sparse masks $\boldsymbol{m}_s$ for the pretraining stage to be a all "1" mask.
Initialize the model's weight as $\boldsymbol{\theta}_0$ and train the weights on $\mathcal{D}_s$ to obtain $\boldsymbol{\theta}_s$.
**for** $i = 1,2,\ldots,N_s$ **do**                                      ▷ IMP at the pre-training stage
    Prune 20% of the smallest-magnitude weights from the non-zero regions of $\boldsymbol{m}_s \odot \boldsymbol{\theta}_s$, by setting the values at corresponding positions to those weights in $\boldsymbol{m}_s$ to "0".
    (Re-)train the sparse weights $\boldsymbol{m}_s \odot \boldsymbol{\theta}_s$ on $\mathcal{D}_s$. Only $\boldsymbol{\theta}_s$ is updated.
**end for**
Initialize the sparse masks at the fine-tuning stage $\boldsymbol{m}_t$ to be all "1" masks and freeze $\boldsymbol{m}_s$.
Initialize model's weight as $\boldsymbol{m}_s \odot \boldsymbol{\theta}_s$, and train on $\mathcal{D}_t$ to obtain $\boldsymbol{m}_s \odot \boldsymbol{\theta}_t$.
**for** $i = 1,2,\ldots,N_t$ **do**                                      ▷ IMP at the fine-tuning stage
    Prune 20% of the smallest-magnitude weights from the non-zero regions of weights $(\boldsymbol{m}_s \odot \boldsymbol{m}_t) \odot \boldsymbol{\theta}_s$, by setting the values at corresponding positions to those weights in $\boldsymbol{m}_t$ to "0".
    (Re-)train the sparse weights $(\boldsymbol{m}_s \odot \boldsymbol{m}_t) \odot \boldsymbol{\theta}_t$ on $\mathcal{D}_t$. Only $\boldsymbol{\theta}_t$ is updated.
**end for**
Obtain the final sparse weights $(\boldsymbol{m}_s \odot \boldsymbol{m}_t) \odot \boldsymbol{\theta}^*$ and return $\boldsymbol{\theta}^*$, $\boldsymbol{m}_s$ and $\boldsymbol{m}_t$.

---

**Formulation**

$$\min_{\boldsymbol{\theta},\boldsymbol{m}_s,\boldsymbol{m}_t} \mathbb{E}_{(\boldsymbol{x}_t,y_t)\sim\mathcal{D}_t} \left[\mathcal{L}_t((\boldsymbol{m}_s \odot \boldsymbol{m}_t) \odot \boldsymbol{\theta},\boldsymbol{x}_t,y_t|\boldsymbol{\theta}^*,\boldsymbol{m}_s^*)\right] + \gamma\mathcal{R}(\boldsymbol{m}_s^* \odot \boldsymbol{m}_t)$$

$$\text{s.t. } \{\boldsymbol{\theta}^*,\boldsymbol{m}_s^*\} = \arg\min_{\boldsymbol{\theta},\boldsymbol{m}_s} \mathbb{E}_{(\boldsymbol{x}_s,y_s)\sim\mathcal{D}_s} \mathcal{L}_s(\boldsymbol{m}_s \odot \boldsymbol{\theta},\boldsymbol{x}_s,y_s).$$

**Interpretation** Let us start from the *lower-level* problem (2) which instantiates the sparsity regularized pre-training stage over $\mathcal{D}_s$: its outputs include the pre-trained weight $\boldsymbol{\theta}^*$ and the corresponding sparse mask $\boldsymbol{m}_s$. Then, the *upper-level* problem (1) depicts the sparsity regularized fine-tuning over $\mathcal{D}_t$, which inherits both $\boldsymbol{\theta}^*$ and $\boldsymbol{m}_s^*$ as its starting point. It continues to modify the weight as well as to evolve another sparse mask $\boldsymbol{m}_t$. Eventually, a sparsity-promoting function $\mathcal{R}$ enforces the total sparsity over the joint mask $\boldsymbol{m}_s \odot \boldsymbol{m}_t$, and the final model weights could be represented as $(\boldsymbol{m}_s \odot \boldsymbol{m}_t) \odot \boldsymbol{\theta}$.

**Lower-level problem** We solve the lower-level problem through a $p$-step SGD unrolling. Let $\boldsymbol{\theta}^{(k)}$ be the model weights, and $\boldsymbol{m}_s^{(k)}$ be the mask for the pretraining stage. The superscript $(k)$ indicates they have been updated on the upper-level for $k$ steps.

$\boldsymbol{\theta}^{(k)}$ and $\boldsymbol{m}_s^{(k)}$ will be the starting points for the lower-level optimization problem. $\boldsymbol{\theta}_l^{(t)}$ and $\boldsymbol{m}_{s,l}^{(t)}$ are the weights and mask, respectively, after being updated for $t$ steps on the lower-level optimization problem (implying $\boldsymbol{\theta}_l^{(0)} = \boldsymbol{\theta}^{(k)}$ and $\boldsymbol{m}_{s,l}^{(0)} = \boldsymbol{m}_s^{(k)}$). The update rules can be written as

$$\boldsymbol{\theta}_l^{(0)} = \boldsymbol{\theta}^{(k)}, \boldsymbol{\theta}_l^{(p)} = \boldsymbol{\theta}_l^{(p-1)} - \lambda_l \nabla_{\boldsymbol{\theta}}\mathcal{L}_s|_{\boldsymbol{\theta}=\boldsymbol{\theta}_l^{(p-1)}}, \tag{3}$$

$$\boldsymbol{m}_{s,l}^{(0)} = \boldsymbol{m}_s^{(k)}, \boldsymbol{m}_{s,l}^{(p)} = \boldsymbol{m}_{s,l}^{(p-1)} - \lambda_{m,l} \nabla_{\boldsymbol{m}}\mathcal{L}_s|_{\boldsymbol{m}=\boldsymbol{m}_{s,l}^{(p-1)}}, \tag{4}$$

where $\lambda_l$ is the learning rate for the model weight $\boldsymbol{\theta}$, and $\lambda_{m,l}$ is the learning rate for the mask $\boldsymbol{m}_{s,l}^{(t)}$ at the lower-level optimization problem.

**Upper-level problem and Sparse Regularization Loss** The upper-level problem is the sum of two losses: a normal training loss $\mathcal{L}_t$ and a sparse regularization loss $\mathcal{R}$ ($\gamma$ is a coefficient).

We first develop update rules for the training loss $\mathcal{L}_t$. The weights $\boldsymbol{\theta}^*(:= \boldsymbol{\theta}_l^{(p)})$ and masks $\boldsymbol{m}_s^*(:= \boldsymbol{m}_{s,l}^{(p)})$ from the lower-level problem after $p$ unroll steps will serve as the initialization of the upper-level problem. We update the model weight $\boldsymbol{\theta}$ and masks at the upper level by applying gradient-based methods (take SGD as an example):

$$\boldsymbol{\theta}^{(k+1)} = \boldsymbol{\theta}^* - \lambda_u \frac{\mathrm{d}\mathcal{L}_t}{\mathrm{d}\boldsymbol{\theta}^*} \tag{5}$$

$$= \boldsymbol{\theta}^* - \lambda_u \Big(\frac{\partial \mathcal{L}_t}{\partial \boldsymbol{\theta}^*} + \frac{\partial \mathcal{L}_t}{\partial \boldsymbol{m}_s^*} \frac{\partial \boldsymbol{m}_s^*}{\partial \boldsymbol{\theta}^*}\Big),$$

where $\lambda_u$ is the learning rate for the weights for the upper-level optimization problem. The gradient on $\boldsymbol{m}_t$ is easy enough: $\frac{\partial \mathcal{L}_t}{\partial \boldsymbol{m}_t}$, while the gradient on $\boldsymbol{m}_s$ is slightly complicated:

$$\frac{\mathrm{d}\mathcal{L}_t}{\mathrm{d}\boldsymbol{m}_s^*} = \frac{\partial \mathcal{L}_t}{\partial \boldsymbol{m}_s^*} + \frac{\partial \mathcal{L}_t}{\partial \boldsymbol{\theta}^*} \frac{\partial \boldsymbol{\theta}^*}{\partial \boldsymbol{m}_s^*}. \tag{6}$$

We expand the latter terms in Eqn. 5 and Eqn. 6 based on the first-order approximation (picking $p = 1$) on the lower-level problem:

$$\frac{\partial \boldsymbol{\theta}^*}{\partial \boldsymbol{m}_s^*} = \frac{\partial(\boldsymbol{\theta}_l^{(0)} - \lambda_l \nabla_{\boldsymbol{\theta}} \mathcal{L}_s)}{\partial(\boldsymbol{m}_{s,l}^{(0)} - \lambda_{m,l} \nabla_{\boldsymbol{m}_s} \mathcal{L}_s)} = \frac{\partial(\boldsymbol{\theta}_l^{(0)} - \lambda_l \nabla_{\boldsymbol{\theta}} \mathcal{L}_s)}{\partial \boldsymbol{\theta}_l^{(0)}} \frac{\partial \boldsymbol{\theta}_l^{(0)}}{\partial(\boldsymbol{m}_{s,l}^{(0)} - \lambda_{m,l} \nabla_{\boldsymbol{m}_s} \mathcal{L}_s)} + \tag{7}$$

$$\frac{\partial(\boldsymbol{\theta}_l^{(0)} - \lambda_l \nabla_{\boldsymbol{\theta}} \mathcal{L}_s)}{\partial \boldsymbol{m}_{s,l}^{(0)}} \frac{\partial \boldsymbol{m}_{s,l}^{(0)}}{\partial(\boldsymbol{m}_{s,l}^{(0)} - \lambda_{m,l} \nabla_{\boldsymbol{m}_s} \mathcal{L}_s)}$$

$$= (\mathrm{I} - \lambda_l \nabla_{\boldsymbol{\theta}}^2 \mathcal{L}_s)(-\lambda_{m,l} \nabla_{\boldsymbol{m}_s \boldsymbol{\theta}} \mathcal{L}_s)^{-1} +$$

$$(-\lambda_l \nabla_{\boldsymbol{m}_s \boldsymbol{\theta}} \mathcal{L}_s)(\mathrm{I} - \lambda_{m,l} \nabla_{\boldsymbol{m}_s}^2 \mathcal{L}_s)^{-1},$$

$$\frac{\partial \boldsymbol{m}_s^*}{\partial \boldsymbol{\theta}^*} = \frac{\partial(\boldsymbol{m}_{s,l}^{(0)} - \lambda_{m,l} \nabla_{\boldsymbol{m}_s} \mathcal{L}_s)}{\partial(\boldsymbol{\theta}_l^{(0)} - \lambda_l \nabla_{\boldsymbol{\theta}} \mathcal{L}_s)} = (\mathrm{I} - \lambda_l \nabla_{\boldsymbol{\theta}}^2 \mathcal{L}_s)^{-1}(-\lambda_{m,l} \nabla_{\boldsymbol{m}_s \boldsymbol{\theta}} \mathcal{L}_s) +$$

$$(-\lambda_l \nabla_{\boldsymbol{m}_s \boldsymbol{\theta}} \mathcal{L}_s)^{-1}(\mathrm{I} - \lambda_{m,l} \nabla_{\boldsymbol{m}_s}^2 \mathcal{L}_s). \tag{8}$$

Further approximations can be made to avoid the matrix inverse and save computation:

$$\frac{\partial \boldsymbol{\theta}^*}{\partial \boldsymbol{m}_s^*} \approx -\lambda_l \nabla_{\boldsymbol{m}_s \boldsymbol{\theta}} \mathcal{L}_s \;, \frac{\partial \boldsymbol{m}_s^*}{\partial \boldsymbol{\theta}^*} \approx -\lambda_{m,l} \nabla_{\boldsymbol{m}_s \boldsymbol{\theta}} \mathcal{L}_s.$$

Based on the rules, $\boldsymbol{m}_s$ and $\boldsymbol{m}_t$ can be optimized by:

$$\hat{\boldsymbol{m}}_t^{(k+1)} = \boldsymbol{m}_t^{(k)} - \lambda_m \frac{\partial \mathcal{L}_t}{\partial \boldsymbol{m}_t}\Big|_{\boldsymbol{m}_t = \boldsymbol{m}_t^{(k)}} \tag{9}$$

$$\hat{\boldsymbol{m}}_s^{(k+1)} = \boldsymbol{m}_s^{(k)} - \lambda_m \frac{\partial \mathcal{L}_t}{\partial \boldsymbol{m}_s} + \lambda_m \lambda_l \frac{\partial \mathcal{L}_t}{\partial \boldsymbol{\theta}^*} \nabla_{\boldsymbol{m}_s \boldsymbol{\theta}} \mathcal{L}_s\Big|_{\boldsymbol{m}_s = \boldsymbol{m}_s^{(k)}}, \tag{10}$$

where the superscript $(k)$ means the steps updated.

We then focus on the latter term. We choose the $\ell_0$ loss (*i.e.* the number of non-zero elements) as the sparse regularizer $\mathcal{R}$, which is not differentiable and difficult to optimize. Therefore, we follow Guo et al. [47] to use the smoothed $\ell_0$ formulation to facilitate differentiable training. Specifically, a gate function $g_\epsilon(\boldsymbol{x}) \coloneqq \frac{\boldsymbol{x}^2}{\boldsymbol{x}^2 + \epsilon}$, where $\epsilon$ is a small positive number, is used to replace the binary masks, which are instead parameterized by $g_\epsilon(\boldsymbol{m}_s)$ and $g_\epsilon(\boldsymbol{m}_t)$. We decay the value of $\epsilon$ every epoch, and the gate function will gradually output only polarized numbers (*i.e.*, 0 and 1). We further apply the proximal-SGD [54] to minimize the $\ell_0$ loss: after we update the $\boldsymbol{m}_s$ and $\boldsymbol{m}_t$ with respect to $\mathcal{L}_t$ by gradient descent-based methods (Eqn. 10), we use the proximal operator to alternatively update each mask. For $\boldsymbol{m}_s$, the formulation can be written as:

$$\mathrm{prox}_{\lambda_m \gamma \mathcal{R}}(\boldsymbol{m}_s^{(k+1)}) =$$

$$\operatorname*{arg\,min}_{\boldsymbol{m}_s} \frac{1}{2} \|\boldsymbol{m}_s^{(k+1)} \odot \hat{\boldsymbol{m}}_t^{(k+1)} - \hat{\boldsymbol{m}}_s^{(k+1)} \odot \hat{\boldsymbol{m}}_t^{(k+1)}\|_2^2 + \lambda_m \gamma \|\boldsymbol{m}_s^{(k+1)} \odot \hat{\boldsymbol{m}}_t^{(k+1)}\|_0.$$

We follow [47] to solve it by relaxing it to the $\ell_1$ norm problem, which has a closed form solution:

$$
m_{s,i} = \begin{cases} \hat{m}_{s,i}^{(k+1)} - \frac{\gamma\lambda_m}{\hat{m}_{t,i}^{(k+1)}}, & \hat{m}_{s,i}^{(k+1)} \geq \frac{\gamma\lambda_m}{\hat{m}_{t,i}^{(k+1)}} \\ \hat{m}_{s,i}^{(k+1)} + \frac{\gamma\lambda_m}{\hat{m}_{t,i}^{(k+1)}}, & \hat{m}_{s,i}^{(k+1)} \leq -\frac{\gamma\lambda_m}{\hat{m}_{t,i}^{(k+1)}} \\ 0, & -\frac{\gamma\lambda_m}{\hat{m}_{t,i}^{(k+1)}} < \hat{m}_{s,i}^{(k+1)} < \frac{\gamma\lambda_m}{\hat{m}_{t,i}^{(k+1)}} \end{cases}, \tag{11}
$$

where $m_{s,i}$ is the $i$-th element in $\boldsymbol{m}_s$ (the same for $m_{t,i}$).

Similarly, we derive the update for $\boldsymbol{m}_t$:

$$
m_{t,i}^{(k+1)} = \begin{cases} \hat{m}_{t,i}^{(k+1)} - \frac{\gamma\lambda_m}{\hat{m}_{s,i}^{(k+1)}}, & \hat{m}_{t,i}^{(k+1)} \geq \frac{\gamma\lambda_m}{\hat{m}_{t,i}^{(k+1)}} \\ \hat{m}_{t,i}^{(k+1)} + \frac{\gamma\lambda_m}{\hat{m}_{s,i}^{(k+1)}}, & \hat{m}_{t,i}^{(k+1)} \leq -\frac{\gamma\lambda_m}{\hat{m}_{t,i}^{(k+1)}} \\ 0, & -\frac{\gamma\lambda_m}{\hat{m}_{s,i}^{(k+1)}} < \hat{m}_{t,i}^{(k+1)} < \frac{\gamma\lambda_m}{\hat{m}_{s,i}^{(k+1)}} \end{cases}. \tag{12}
$$

Finally, we combine all these components into Algorithm 2.

---

**Algorithm 2** Solving *Bi-RPT*

---

**Input:** Initialization weights $\boldsymbol{\theta}_0$, training loss functions for two stages $\mathcal{L}_s$ and $\mathcal{L}_t$, low-quality pretraining dataset $\mathcal{D}_s$, high-quality fine-tuning dataset $\mathcal{D}_t$, number of steps for gradient unroll $p$.
**Output:** Trained model weights $\boldsymbol{\theta}$, sparse masks $\boldsymbol{m}_s$ and $\boldsymbol{m}_t$.
Train $\boldsymbol{\theta}_0$ on $\mathcal{D}_s$ to get weights $\boldsymbol{\theta}$.
**while** not converged **do**
    Given the fixed $\boldsymbol{m}_s$, update the weights $\boldsymbol{\theta}$ on $\mathcal{D}_s$ by gradient unrolling (Eqn. 3)
    Update the weights $\boldsymbol{\theta}$ by Eqn. 5
    Update the masks $\boldsymbol{m}_s$ and $\boldsymbol{m}_t$ by Eqn. 10.
    Update the masks $\boldsymbol{m}_s$ and $\boldsymbol{m}_t$ by Eqn. 11 and Eqn. 12.
**end while**

---

# B. More Experiments Details and Results

## B.1. Baselines, Hyperparameters and Architectures.

We list the hyper-parameters we used for all the baselines in this section. **General Settings.** When pre-training the models on $\mathcal{D}_s$ (ImageNet and ImageNet-C), we use the SGD optimizer and a learning rate of $4 \times 10^{-1}$. We linearly warm-up the learning rate within 5 epochs, and then decay it by 10 for every 30 epochs. Models are pretrained for 95 epochs on $\mathcal{D}_s$, with a batch size of 1024. On $\mathcal{D}_t$, *i.e.*, CUB-200 and CUB-200 (10-shot), we set the initial learning rate as $1 \times 10^{-3}$. The learning rate is decayed by 10 every 30 epochs, and the model is trained for 90 epochs with a batch size of 64.

For Hand-Tune, we train the models with 95 epochs from scratch on $\mathcal{D}_s$ to get a *densely* pretrained model. The number of training epochs is reduced to 45 after the model is pretrained. Following the pretraining stage, we continue to transfer the model on $\mathcal{D}_t$ using the above hyper-parameters. The number of epochs is reduced to 45 after we prune the weights.

For No-Pretraining, we train the model using an initial learning rate of $1 \times 10^{-2}$ and a batch size of 64. For Mix-Training, as the number of classes is different for ImageNet and CUB-200, we use two fully-connected layers on top the normal ResNet-18 backbone, and train them simultaneously. We sample batches from the two domains ($\mathcal{D}_s$ and $\mathcal{D}_t$) using the same batch size of 64. The initial learning rate for these methods is $1 \times 10^{-2}$ and is decayed by 10 every 30 epochs.

For *Bi-RPT*, we follow the same learning rate settings and introduce two additional hyper-parameters. The learning rate for the lower-level problem ($\lambda_l$) is $1 \times 10^{-3}$, the same as the learning rate for the upper-level problem ($\lambda_u$). The value of $\gamma$ is set to $1 \times 10^{-4}$, which is determined through

ablation studies in Table 9. The value of $\lambda_m$ is set to $3.5$, which is also determined through ablation studies in Table 10.

For experiments on **X-Yield**, we pretrain the ML predictor on the simulation data for $10$ epochs. During the pretraining, we use the Adam optimizer [55] with an initial learning rate of $1 \times 10^{-4}$ and a cosine annealing schedule [56]. For the transfer stage, we fine-tune the pretrained model on the experimental data for $90$ epochs. We use the SGD optimizer with an initial learning rate of $1 \times 10^{-3}$. We also decay the learning rate by $10$ for every $30$ epochs. The batch sizes for pretraining and fine-tuning are $16$ and $4$, respectively.

The ML predictor we use is a convolutional neural network. It consists of 3 convolutional layers, each of which has a kernel size of 3, followed by Batch Normalization [52] and ReLU [53] activation. A multi-layer perceptron is appended after the convolutional neural network to generate the final prediction.

## B.2. Performance of Hand-Tune Under Different Levels of Sparsity.

We report the performance of the Hand-Tune method under different levels of sparsity. We conduct experiments with $N_s = \{0,1,2,3,4,5\}$ and $N_t = \{0,1,2,3,4\}$, resulting in sparsity levels during pretraining of $\{0.00\%, 20.00\%, 36.00\%, 48.80\%, 59.04\%, 67.23\%\}$ and sparsity levels during fine-tuning of $\{0.00\%, 20.00\%, 36.00\%, 48.80\%, 59.04\%\}$. We conduct experiments over all combinations of pretraining and transfer pruning rounds. More specifically, we first perform IMP on $\mathcal{D}_s$ for $N_s$ rounds, and continue to perform IMP on $\mathcal{D}_t$ for another $N_t$ rounds. The experimental results over various source and target combinations are shown in Tables 4 - 7. Note that all models are evaluated on the testing samples in $\mathcal{D}_t$.

From this series of tables we observe that: (1) sparsity during pretraining helps improve model performance on $\mathcal{D}_t$ after fine-tuning, and the performance gain is larger when $\mathcal{D}_s$ contains more noise and has larger domain shifts; (2) sparsity during fine-tuning also benefits performance after fine-tuning, and the improvement is more significant when the $\mathcal{D}_t$ is more "data-scarce"; (3) the optimal sparsity levels for the two stages vary for different combinations of pretrain and transfer domains, highlighting the importance of choosing the correct pruning rounds for both stages.

Table 4: Test accuracy of fine-tuned ResNet-18 on CUB-200 after pretrained on ImageNet, under different levels of sparsity at pretraining and sparsity at transfer.

| Sparsity At Transfer | Sparsity At Pretraining | | | | | |
|---|---|---|---|---|---|---|
| | 0.00% | 20.00% | 36.00% | 48.80% | 59.04% | 67.23% |
| 0.00% | 74.16% | **76.01%** | 75.77% | 75.87% | 74.99% | 74.35% |
| 20.00% | 74.15% | 75.54% | 75.82% | 75.98% | 74.73% | 74.46% |
| 36.00% | 74.13% | 75.08% | 75.56% | 75.73% | 74.06% | 73.94% |
| 48.80% | 73.84% | 74.01% | 74.56% | 74.46% | 72.37% | 72.16% |
| 59.04% | 73.61% | 73.77% | 73.61% | 72.89% | 70.66% | 70.56% |

Table 5: Test accuracy of fine-tuned ResNet-18 on CUB-200 after pretrained on ImageNet-C, under different levels of sparsity at pretraining and sparsity at transfer.

| Sparsity At Transfer | Sparsity At Pretraining | | | | | |
|---|---|---|---|---|---|---|
| | 0.00% | 20.00% | 36.00% | 48.80% | 59.04% | 67.23% |
| 0.00% | 71.59% | 71.89% | 73.44% | 73.70% | 73.63% | 73.52% |
| 20.00% | 71.83% | 72.44% | 73.97% | **74.01%** | 73.52% | 73.39% |
| 36.00% | 71.68% | 72.80% | 73.46% | 73.47% | 72.95% | 72.70% |
| 48.80% | 71.13% | 71.87% | 72.14% | 72.40% | 71.87% | 71.28% |
| 59.04% | 69.26% | 70.31% | 70.61% | 70.73% | 70.11% | 69.38% |

Table 6: Test accuracy of fine-tuned ResNet-18 on CUB-200 (10-shot) after pretrained on ImageNet, under different levels of sparsity at pretraining and sparsity at transfer.

| Sparsity At Transfer | Sparsity At Pretraining | | | | | |
|---|---|---|---|---|---|---|
| | 0.00% | 20.00% | 36.00% | 48.80% | 59.04% | 67.23% |
| 0.00% | 38.66% | 35.88% | 38.30% | 39.23% | 40.73% | 39.14% |
| 20.00% | 38.90% | 36.56% | 38.66% | 40.14% | **40.78%** | 39.33% |
| 36.00% | 38.42% | 36.31% | 38.95% | 40.14% | 40.32% | 38.97% |
| 48.80% | 38.13% | 35.23% | 37.80% | 38.02% | 38.37% | 35.69% |
| 59.04% | 37.02% | 33.21% | 35.83% | 35.54% | 35.55% | 32.78% |

Table 7: Test accuracy of fine-tuned ResNet-18 on CUB-200 (10-shot) after pretrained on ImageNet-C, under different levels of sparsity at pretraining and sparsity at transfer.

| Sparsity At Transfer | Sparsity At Pretraining | | | | | |
|---|---|---|---|---|---|---|
| | 0.00% | 20.00% | 36.00% | 48.80% | 59.04% | 67.23% |
| 0.00% | 32.14% | 34.29% | 38.07% | 36.12% | 38.21% | 36.85% |
| 20.00% | 32.53% | 35.99% | 39.63% | 37.92% | **39.94%** | 37.66% |
| 36.00% | 32.52% | 36.07% | 38.99% | 38.56% | 39.07% | 36.95% |
| 48.80% | 31.69% | 34.85% | 37.59% | 36.45% | 36.69% | 34.79% |
| 59.04% | 30.64% | 32.55% | 34.79% | 33.31% | 33.72% | 32.64% |

## B.3. HEA Data Representations

The raw data consists of each alloy's composition and the temperature at which the experiment is conducted, resulting in 12-dimensional vectors. We transform these vectors into 2D images using the process outlined in Figure 4. Given the composition of an alloy, the periodic table representation (PTR) sets the percentage of each element into a specific position according to its position in the periodic table and the randomized periodic table representation (RPTR) sets the percentage of each element with a pre-defined shuffled periodic table [51]. In our experiments, we use the RPTR to map values in a more balanced way [51].

## B.4. Ablations on Image Data

**Effects of sparsity at two stages.** We conduct a set of ablation experiments to study the effects of two-stage sparse masks in the *Bi-RPT* formulation on ResNet-18 (pretrained by ImageNet-C, fine-tuned on CUB-200). We compare against three baselines: fixing $m_s$, fixing $m_t$, and fixing both of them. The performance comparison is shown in Table 8, where we can see that learning masks at both stage yields the highest performance.

**Effects of $\gamma$.** We conduct a set of ablation experiments to study the effects of different $\gamma$ on ResNet-18 (pretrained on ImageNet-C, fine-tuned on CUB-200).We vary $\gamma$ within $\{0.5,1,2,3\} \times 10^{-4}$ and present the results in Table 9. We see that $\gamma = 1 \times 10^{-4}$ yields the best performance.

**Effects of learning rates.** We conduct a set of ablation experiments on ResNet-18 (pretrained on ImageNet-C, fine-tuned on CUB-200) to study the effects of learning rate on $m_s$ and $m_t$. The learning rates we study in this ablation experiments are $\{2.5,3.0,3.5,4.0,4.5,5.0\}$. We present the test accuracies in Table 10 and observe that *Bi-RPT* consistently outperform baselines ($74.01\%$) within a wide range of $\lambda_m$.

## B.5. Visualization

In Figure 5, we present visualizations of the sparsity patterns learned by *Bi-RPT* at two distinct stages.

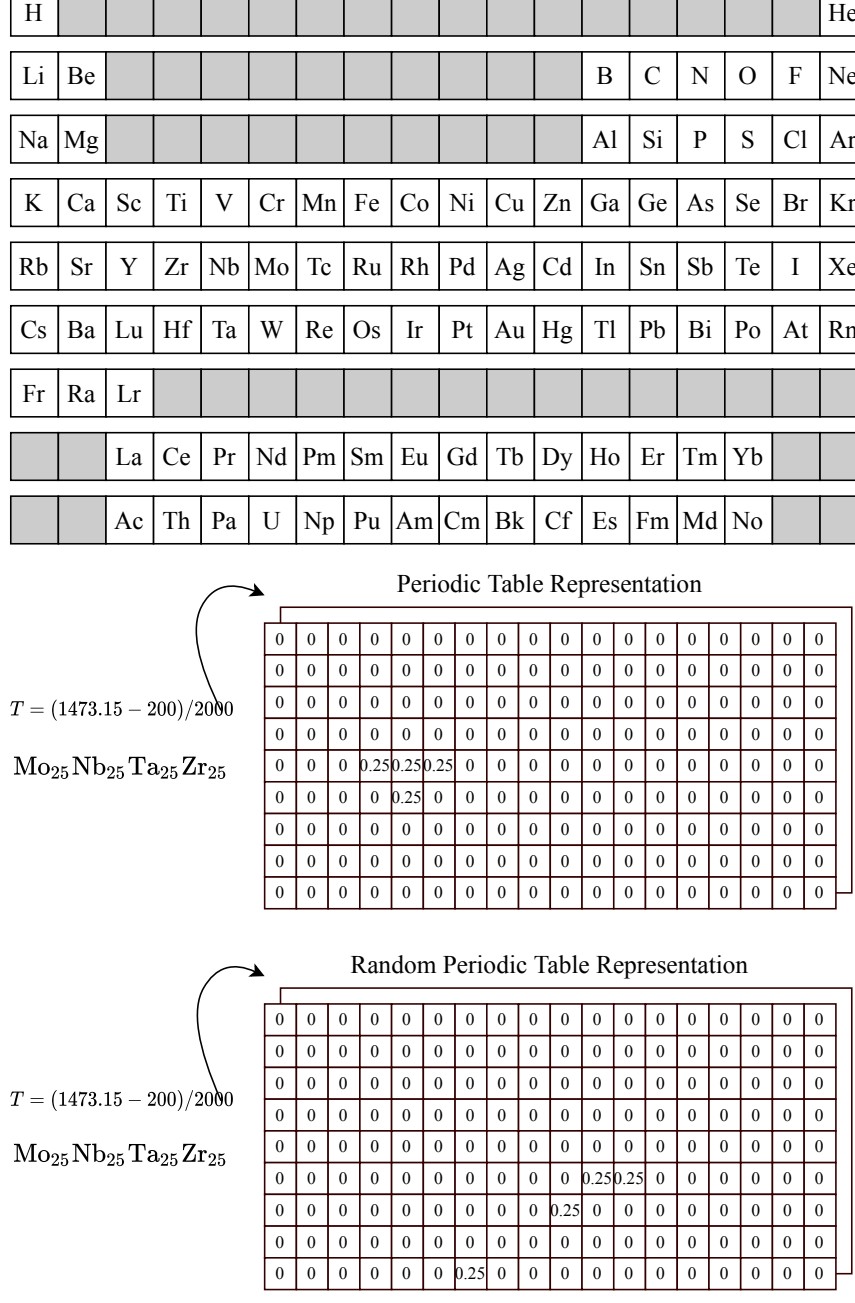

Figure 4: The pipeline for converting a raw input into a pseudoimage. The temperature is embedded as the value of the second channel.

## B.6. Cross Validation

## B.7. Uncertainty Quantification

We provide additional analysis of uncertain quantification. We ensemble ten models trained with *Bi-RPT* and pretrain-and-transfer (PT) methods by averaging their predictions [57], and calculate the standard deviation of the predictions as the uncertainty. The results after ensemble are shown in Table 12.

Table 8: Ablation study on different sparse masks on image data. "Fixed" means the value of $h$ unchanged. We study the combination of pretraining on ImageNet-C and transferring to Birds.

| Mask Type | Test Accuracy |
|---|---|
| Fixed $m_s$ and $m_t$ | 71.58% |
| Fixed $m_s$ | 72.09% |
| Fixed $m_t$ | 75.53% |
| *Bi-RPT* | 76.29% |

Table 9: Ablation study on the effects of different $\gamma$ on the image data. Test accuracy of fine-tuned ResNet-18 on CUB-200 after pretrained on ImageNet-C is reported.

| $\gamma$ | Test Accuracy |
|---|---|
| $0.5 \times 10^{-4}$ | 72.32% |
| $1 \times 10^{-4}$ | 76.29% |
| $2 \times 10^{-4}$ | 65.42% |
| $3 \times 10^{-4}$ | 52.59% |

We see that the ensemble of sparse models produces more accurate predictions overall than the baseline ensemble of pretrain-and-transfer. Furthermore, we found that the uncertainty and prediction error associated with the use of the sparse model ensemble are moderately positively correlated, with a value of approximately $0.15$. In contrast, the correlation between the uncertainty and prediction error derived from using the ensemble of dense models is weaker, with a value of around $-0.24$.

### B.8. Comparison with Few-shot Methods

For these experiments, we first pre-train models on mini-ImageNet and subsequently transfer them to CUB. We compare the performance of *Bi-RPT* with TIM [50], a state-of-the-art approach [58] to few-shot classification with domain shifts. The original TIM method does not tune the backbone. Therefore we conduct experiments where the backbones are fine-tuned (called TIM + Backbone Tuning) for a fair comparison. We resize the input image of CUB to $224 \times 224$ to ensure a fair comparison with TIM. The average accuracy from 100 runs is shown in Table 13. *Bi-RPT* out-performs TIM + Backbone Tuning by $3.13\%/2.14\%/5.76\%$ and out-performs TIM by $2.12\%/2.20\%/6.07\%$ in the 5/10/20-shot setting, respectively.

## C. More Details on Dataset

### C.1. Dataset Comparison

We have provided a comparison between different relevant datasets in Table 14. We elaborate more on the differences:

1. Maresca et al. [4] have only sparse data from the Mo-Nb-Ta-V-W element family.

2. Lee et al. [59] have released a database of the predicted YS of 10 million alloys from the Al-Cr-Mo-Nb-Ta-V-W-Hf-Ti-Zr family at 1300 K. Our dataset contains alloys from Al-Cr-**Fe**-Mo-Nb-Ta-V-W-Hf-Ti-Zr family at temperatures from **300 K** to **2500 K**. Our simulation data are significantly larger (over 3 billion samples) and this dataset is available in its entirety, while only 100K are included for training the ML models in this study.

3. Borg et al. [7] compile experimental data from published materials science articles since 2004. The dataset contains 630 samples with different crystal structures. Our experimental dataset is a subset of Borg et al [7] focusing on alloys with BCC structures only.

Table 10: Ablation study on the effects of different learning rate on $\boldsymbol{m}_s$ and $\boldsymbol{m}_t$ on image data. Test accuracy of fine-tuned ResNet-18 on CUB-200 after pretrained on ImageNet-C is reported.

| $\lambda_m$ | Test Accuracy |
|---|---|
| 2.5 | 72.88% |
| 3.0 | 75.73% |
| 3.5 | 76.29% |
| 4.0 | 75.94% |
| 4.5 | 73.69% |
| 5.0 | 73.34% |

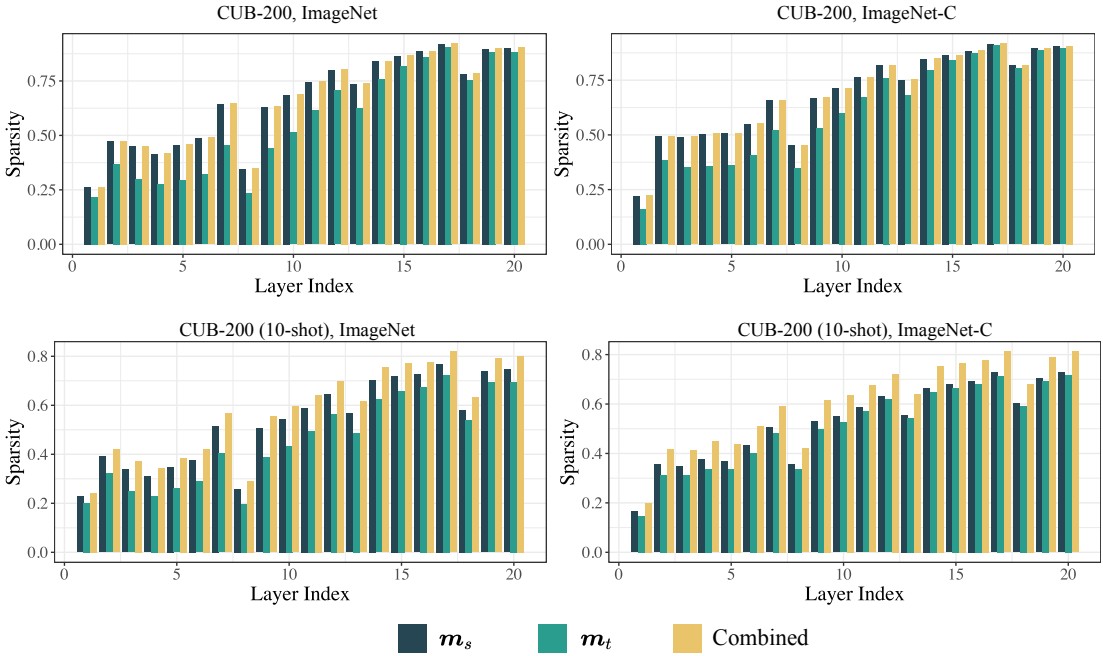

Figure 5: Layerwise sparsity learned by *Bi-RPT* on CUB-200 and Birds-S with ImageNet and ImageNet-C pretraining. We report the sparsity level of the two masks, as well as their combined sparsity (note that *Bi-RPT* allows for the two masks to partially overlap).

Table 11: Classification and regression performance with 10-fold cross-validation. We report both the mean and the 95% confidence interval.

| | Classification (Accuracy) | Regression (MSE) |
|---|---|---|
| No Pretraining | $67.50 \pm 5.16\%$ | $0.226 \pm 0.027$ |
| Pretrain-and-transfer | $82.09 \pm 3.86\%$ | $0.206 \pm 0.026$ |
| *Bi-RPT* | $\mathbf{82.50 \pm 2.93}\%$ | $\mathbf{0.068 \pm 0.009}$ |

Table 12: Uncertainty estimation calculated by ensembling independently trained models. We study two methods: pretrain-and-transfer (PT) and *Bi-RPT*. The results after ensemble are reported as PT-Ensemble and *Bi-RPT*-Ensemble, respectively. The estimated uncertainty is reported in brackets.

| Alloys | Temperature (K) | Predicted Yield Stress (GPa) | | | | Experimental (GPa) |
|---|---|---|---|---|---|---|
| | | *Bi-RPT* | *Bi-RPT*-Ensemble | PT | PT-Ensemble | |
| MoNbTaTi | 293.15 | 1.078 | **1.158 (0.083)** | 1.062 | 1.054 (0.011) | 1.210 |
| | 473.15 | 0.965 | 1.046 (0.087) | **0.902** | 0.908 (0.015) | 0.868 |
| | 673.15 | 0.746 | 0.850 (0.085) | **0.731** | 0.740 (0.026) | 0.685 |
| | 873.15 | 0.508 | 0.674 (0.103) | **0.584** | 0.604 (0.021) | 0.593 |
| | 1273.15 | 0.425 | 0.482 (0.088) | 0.488 | **0.501 (0.018)** | 0.539 |
| MoNbTaTiW | 298.15 | **1.268** | **1.268 (0.098)** | 1.068 | 1.062 (0.011) | 1.399 |
| | 873.15 | **0.677** | 0.798 (0.102) | 0.607 | 0.624 (0.022) | 0.689 |
| | 1073.15 | 0.618 | **0.681 (0.111)** | 0.523 | 0.528 (0.013) | 0.674 |
| | 1273.15 | 0.536 | **0.567 (0.124)** | 0.486 | 0.496 (0.017) | 0.620 |
| HfMoNbTaTiZr | 296.15 | **1.527** | 1.392 (0.122) | 1.132 | 1.142 (0.021) | 1.515 |
| | 873.15 | 0.861 | **0.864 (0.098)** | 0.685 | 0.698 (0.017) | 0.973 |
| | 1073.15 | **0.762** | 0.747 (0.105) | 0.612 | 0.624 (0.022) | 0.791 |
| | 1273.15 | **0.662** | 0.646 (0.134) | 0.573 | 0.587 (0.022) | 0.753 |

Table 13: Comparison of *Bi-RPT* against few-shot learning methods. The experiments are repeated 100 times.

| Methods | 5-way 10-shot | 10-way 10-shot | 20-way 10-shot |
|---|---|---|---|
| TIM | 75.35% | 62.31% | 50.76% |
| TIM + Backbone Tuning | 74.34% | 62.37% | 51.07% |
| *Bi-RPT* | 77.47% | 64.51% | 56.83% |

Table 14: Comparison between different datasets.

| Dataset | Alloy Family | Number of data points | Temperature |
|---|---|---|---|
| Maresca et al. [4] | Mo-Nb-Ta-V-W | Sparse | N/A |
| Lee et al. [59] | Al-Cr-Mo-Nb-Ta-V-W-Hf-Ti-Zr | 10 million | 1300 K |
| Borg et al. [7] | N/A | 630 | N/A |
| **X-Yield** | Al-Cr-Fe-Mo-Nb-Ta-V-W-Hf-Ti-Zr | 3 billion | 300 K - 2500 K |

