# OpenReview forum: "Cross-Quality Few-Shot Transfer for Alloy Yield Strength Prediction: A New Materials Science Benchmark and A Sparsity-Oriented Optimization Framework"
_CPAL.cc/2024/Conference — CPAL 2024 (Proceedings Track) Oral_

### Official Review · Reviewer_Kp6T · 2023-10-04
**Interesting, but needs a few more baselines**

**Rating:** 6
**Confidence:** 4

**Review:**

This paper considers the very challenging scenario in material science where one hopes to obtain a predictor of yield strength (YS) of high-entropy alloys (HEAs) when high quality real-world measurements are very scarce and simulation results are not very accurate.

The paper makes 2 main contributions:
1. The authors compiled a new benchmark dataset “X-Yield” that represents the scenario described above.

2. The authors proposed a new sparsity-regularized training scheme (Bi-RPT) which achieves better results than plain pre-training under the setup of strong domain gap and huge dataset size difference between pre-training and fine-tuning.

Overall, the paper is well written and easy to follow. The experimental results are clear and convincing. I do have a few questions and suggestions to the authors which I will detail below:
It is interesting that the proposed Bi-RPT method can improve the transfer performance. My understanding is that this method outperforms baseline pretrain-and-transfer only under strong domain gap settings. From a scientific point of view, a proof-of-concept experiment needs to be included to show a setting where Bi-RPT doesn’t help. For example, when transferring from ImageNet to ImageNetV2, sparsity regularization would likely underperform plain pretrain-transfer. If otherwise, this method should be considered to be a better transfer learning method in general.
From the experimental results in Table 1, it seems that sparsity at the transfer stage doesn’t change the result significantly. Pre-training sparsity helps the result a little bit, and allowing optimization of pre-training sparsity masks on fine-tuning data provides additional gains. This still holds in Table 4,5,6,7 in supplementary materials, although sometimes tuning the transfer sparsity helps, the difference is very minor. Could it be possible that not using the transfer mask in Bi-RPT produces identical results and the method can thus be simplified？
When applying Bi-RPT to X-Yield dataset, the authors transformed the composition of the material and the temperature parameter to a pseudo-image, then used the same image classification Convnet to solve the problem. My concern is that this method is unnecessarily complex, given the material-science problem considered is mostly low-dimensional, could it be that classical regression method like LASSO, or simple MLP works equally well or even better? Since in that setting, similar sparsity constraints can be applied, and the number of parameters is much less. Either way, it would be nicer to include some baseline comparisons to simpler methods to show the necessity of employing a deep CNN.

I’ll update my score if the questions above are sufficiently addressed by the authors.

Disclaimer: I have no expertise in material science and none of my evaluation is based on the material science significance of this paper.

---

### Official Review · Reviewer_73Cr · 2023-10-07
**This paper studies the discovery of high-entropy alloys (HEAs) with high yield strength (YS) in the material science with transfer learning. Because of the lack a comprehensive dataset and the time-consuming process for experimental data, the authors construct a new X-Y field dataset that consists of the majority of simulated data and minority of experimental data.  Moreover, since the traditional transfer learning methods does not perform well on these datasets, the authors propose a sparsity regularizer for both pretraining stage on simulated data and transferring stage on experimental data. Furthermore, the authors incorporate a bi-level optimization framework to jointly learn the optimal masks and the allocation of sparsity for both stages.**

**Rating:** 7
**Confidence:** 4

**Review:**

Pros:
1. Applying the machine learning method to the material science is an interesting and important area.
2. A new benchmark for refractory HEA yield strength prediction is schedule to be public, which may inspire the investigation of new ML technology in the material science field.
3. The bi-level framework has a solid theoretical foundation.
4. This paper is well-written.

Cons:
1. the motivation of incorporation of sparsity is unclear. Although the authors claim that the use of sparse regularizers enhances the robustness and transferability, there is no related previous works in the line 198-202 to support such claim. Moreover, the incorporation of sparsity/pruning reduce the power of the networks, why it will be helpful for the performance in turn? Is this because the networks are easy to overfit on the simulated data?

In general, this paper is well-written and logically structured. The datasets may fill a missing gap in the material science domain and the sparsity regularizer with bi-level optimization is novel and solid. Therefore, I think it is qualified for the acceptance.

---

### Official Review · Reviewer_6ADX · 2023-10-16
**Interesting topic, good writing, may need more comparison as a ML paper**

**Rating:** 6
**Confidence:** 3

**Review:**

The paper is motivated by the time consuming nature of discovering high entropy alloys (HEAs) with high yield strength (YS). Based on the difficulty, the authors try to use ML tools to estimate the YS of alloys directly. To prepare the training dataset, the authors combine a large quantity of simulation data and scarce real data. The authors then propose a novel approach, termed Bi-RPT to deal with the domain gap between the simulation data and real data. Empirical results validate the effectiveness of the proposed method over several baselines.

### Strength
The paper is well-motivated and well-written, and the proposed methods are introduced in detail with derivations.

Both the problem setting and the proposed method are interesting and novel at least to the reviewer's concern

### Weaknesses
The reviewer believes the paper in broad falls within the few-shot learning domain which is a well-estabilished field with plenty of prior work , and since this is an ML conference, when proposing a method, one application may not be enough to demonstrate the universal applicability. Hence the reviewer would suggest comparing the proposed method with more prior work and conducting the experiments on more datasets beyond the YS of alloy.

---

### Meta-Review · Area_Chair_bojB · 2023-11-15

**Recommendation:** Accept (Poster)
**Confidence:** 5

**Metareview:**

This paper has made nice contributions for Alloy Yield Strength Prediction. First, it establishes a new benchmark, which fills the gap in existing literature and would be useful for future study. Second, it considers the domain gap between simulation data and experimental data and proposes a sparsity-oriented bilevel optimization methods to prune the network learned from the source data and also the one fine tuned on the target domain.  All reviewers think the paper is well written. The authors have tried to address the concerns raised in comments. Based on these, I will recommend acceptance.

---

### Decision · Program_Chairs · 2023-11-19

**Decision:**

Accept (Oral)

**Comment:**

All reviewers and AC agreed that the paper is of high quality. This paper has made nice contributions for Alloy Yield Strength Prediction. First, it establishes a new benchmark, which fills the gap in existing literature and would be useful for future study. Second, it considers the domain gap between simulation data and experimental data and proposes a sparsity-oriented bilevel optimization methods to prune the network learned from the source data and also the one fine tuned on the target domain.

The action PC chair for this paper is Qing Qu, who made the decision after carefully reading the paper as well as the comments by all reviewers and AC. The decision is agreed upon by all PC chairs.